# Expression Quantitative Trait Locus Study of Non-Syndromic Cleft Lip with or without Cleft Palate GWAS Variants in Lip Tissues

**DOI:** 10.3390/cells11203281

**Published:** 2022-10-18

**Authors:** Xiaofeng Li, Yu Tian, Ling Qiu, Shu Lou, Guirong Zhu, Yue Gao, Lan Ma, Yongchu Pan

**Affiliations:** 1Jiangsu Province Key Laboratory of Oral Diseases, Nanjing 210000, China; 2Jiangsu Province Engineering Research Center of Stomatologica Translational Medicine, Nanjing 210000, China; 3Department of Orthodontics, The Affiliated Stomatological Hospital of Nanjing Medical University, Nanjing 210000, China

**Keywords:** cleft lip, cleft palate, genetic variation, eQTL, association study

## Abstract

Non-syndromic cleft lip with or without cleft palate (NSCL/P) is a complex disease with a strong genetic component. More than 40 loci have been identified to be associated with the risk of NSCL/P by genome-wide association studies (GWASs), but the majority of these variants are mapped to non-coding regions of the genome. Expression quantitative trait locus (eQTL) studies have increasingly been integrated with GWASs to identify target genes for these non-coding variants. In this study, we generated a unique, lip-specific eQTL dataset from 40 NSCL/P patients. A total of 5158 eQTL SNPs (eSNPs) -689 eQTL genes were identified after multiple corrections. Then, we integrated nominal eQTL SNPs with NSCL/P risk SNPs and identified 243 variants associated with the expression of 18 genes in lip tissues. Functional annotation analysis indicated that these risk eSNPs were significantly enriched in transcription regulation and chromatin open regions on the genome. These susceptible genes were enriched in cell fate determination, the pluripotency of stem cells, and Wnt signaling pathways. Finally, 8 of the 18 susceptible genes were differentially expressed in NSCL/P case-control studies. In summary, we have generated a unique lip-specific eQTL resource and identified multiple associations for NSCL/P risk loci, which should inform functional studies of NSCL/P biology.

## 1. Introduction

Non-syndromic cleft lip with or without cleft palate (NSCL/P) is one of the most common birth defects characterized by a cleft in the lip or palate, with an incidence of 1:1000 worldwide [1]. Environmental exposures, such as malnutrition and smoking during pregnancy, have been shown to play a role in the diseases [2]. Apart from these effects, it has been proven that NSCL/P has a compelling genetic predisposition through twin and familial clustering studies [3]. More than 25% to 35% of cleft lip and palate cases had a family history.

Genome-wide association studies (GWASs) have been widely conducted in the analysis and prediction of disease pathogenesis. So far, more than 40 risk loci have been identified through GWASs [4,5,6,7]. However, GWASs do not necessarily determine causal variants and genes. The majority of the association signals are localized in non-coding regions of the genome, which is a great challenge for biological interpretation [8,9]. In addition, it has been determined that casual genetic variants have usually been mapped to regulatory regions in disease-associated cell types or tissues, altering the expression of downstream target genes. Expression quantitative trait locus (eQTL) analyses, which associate genetic variants with gene expression, have increasingly been integrated with GWASs to identify disease-susceptibility genes [10,11,12,13]. However, many studies have used eQTL data non-specified for NSCL/P, with the exception of a study using the orbicularis oris muscle of NSCL/P-affected individuals [14], there is a lack of eQTL data that is specific for NSCL/P.

To identify susceptibility genes of NSCL/P, we generated an eQTL dataset using 40 lip tissues that were obtained from NSCL/P patients. We combined this information with published GWAS data to identify genetic regulatory patterns of NSCL/P and to gain a better understanding of its biological mechanisms.

## 2. Materials and Methods

### 2.1. Subjects

We obtained redundant lip tissues and venous blood samples from 40 NSCL/P patients, which was approved by the Institutional Review Board of Nanjing Medical University (NJMUERC [2008] No. 20). All patients signed informed consents. All patients were clinically assessed by an experienced oral surgeon to ensure that individuals with other congenital deformities or syndromic orofacial clefts were excluded. The basic information of the participants is shown in Appendix A.

### 2.2. RNA Sequencing and Genotype Data

The total RNA from these lip tissues was extracted using the TRIzol reagent (Invitrogen, Carlsbad, CA, USA). The quality control of RNA was determined using a NanoDrop™ spectrophotometer (Thermo Fisher Scientific, Waltham, MA, USA) and 1% agarose electrophoresis. Sequencing libraries were generated using Nova6000 (Illumina, San Diego, CA, USA) by Genergy Bio (Shanghai, China), following the manufacturer’s recommendations. The transcript per million (TPM) values were used to normalize the expression levels of the genes.

Genotyping of the 40 patients was conducted via the Affymetrix Axiom Genome-Wide CHB1 and CHB2 Array by the CapitalBio Corporation. The imputation of ungenotyped SNPs was calculated using IMPUTE2 [15] from the 1000 Genomes Project. Basic quality control was completed by removing the SNPs that did not meet the locating conditions in autosomal chromosomes, having a call rate <95%, or having a genotype distribution in the controls that deviated from Hardy–Weinberg equilibrium (HWE; *p* < 1.0 × 10^−5^).

### 2.3. eQTL Analysis

eQTL identification was conducted using gene expression as the outcome, SNP dosage as the independent variable, and gender/age as the covariates. The MatrixEQTL [16] package is a toolset implemented in R to identify eQTLs, and detects the association between genotypes and gene expression with a linear regression model. We used this algorithm to test, on a large-scale basis, the associations in our data. Correction for multiple testing was calculated for each of the association tests between SNPs and the transcription start site (TSS), within 1 megabase (MB).

### 2.4. Integrative Analysis of eQTL and GWAS Data

NSCL/P risk SNPs were retrieved from the GWAS catalog database [17], which included 70 independent tag SNPs. In addition, the linkage disequilibrium analysis, based on the 1000 Genomes Project, was conducted to identify all susceptibility SNPs. A direct overlap was performed between NSCL/P risk SNPs and eQTL SNPs. Gene Ontology (GO) and Kyoto Encyclopedia of Genes and Genomes (KEGG) pathway analyses of the eGene regulated by risk SNPs were determined using KOBAS-i [18]. We selected the top 6 pathways or biological processes with the lowest FDR.

### 2.5. Histone Mark and Transcription Factor Binding Site Enrichment Analysis

Histone and transcription factor modifications suggest the potential regulatory regions of the genome. We used this information to ascertain which regions the eSNPs were enriched in and to interpret the latent functions of these eSNPs.

Histone modification data of the embryonic craniofacial tissues were retrieved from GSE97752 [19]. It is ideal to use ChIP-seq data from lip tissues for this enrichment analysis. However, due to the limited size of the lip samples, we failed to create more specific ChIP-seq data from the lip tissues. Given that NSCL/P is a congenital craniofacial malformation and resulted from disturbed processes during embryonic development, the transcription factor ChIP-seq data from ESCs (embryonic stem cells) were retrieved from Cistromes (http://cistrome.org/db) and applied for the enrichment analysis. GREGOR [20] (Genomic Regulatory Elements and Gwas Overlap algorithm) v1.4.0, implemented in Perl, was used to assess eSNP enrichment scores in the above functional genomic regions.

### 2.6. Differentially Expression Analysis

To determine whether candidate genes were differentially expressed in NSCL/P pathogenesis, we retrieved and downloaded two datasets, GSE42589 [21] and GSE85748 [14], which preserved gene expression profiles from the GEO database. GSE42589 included 7 NSCL/P patients and 6 healthy individuals, and GSE85748 contained 42 NSCL/P-affected individuals and 4 healthy controls. The candidate gene expression levels between the cases and controls were assessed using Student’s *t*-test. A *p*-value less than 0.05 was considered to be statistically significant.

## 3. Results

### 3.1. eQTL Analysis of 40 Lip Tissues from NSCL/P Patients

Forty lip samples were obtained from the Chinese Han population; sample characteristics, including demographics and subtype classifications, are listed in Appendix A. We determined the association between 2.4 million genetic variants and the expression of 29,952 genes using MatrixeQTL (Figure 1A). The analysis was limited to cis-eQTLs in which SNPs were within 1 megabase (Mb) distance from the transcription start site of the genes, and we identified 689 eGenes and 5158 eQTL SNPs (eSNPs) (FDR < 0.05) (Figure 1B, Appendix A). Then, we performed GO enrichment analyses on these eGenes and found that they were enriched in transcription regulation, G protein-coupled receptor signaling, keratinization, and olfactory perception pathways (Appendix A), indicating that these eGenes play vital roles in cell energy synthesis, gene expression, and normal craniofacial function development.

### 3.2. Integrative Analysis of Lip eQTL and NSCL/P Risk SNPs

From the GWAS catalog database, we screened out 70 NSCL/P risk SNPs, which met the condition that *p* < 5 × 10^−8^ (Figure 2A). Then, we performed linkage disequilibrium analyses and identified 2832 tagging SNPs. Together, a total of 2902 risk SNPs were collected for NSCL/P. After overlapping these SNPs and nominal eSNPs (*p* < 0.05), 243 risk-eSNPs with 303 associations were obtained. These SNPs, which were all located within 250 kb of the eGene TSS, regulated 18 genes, such as *TAF1B*, *WNT9B*, and *RAD54B* (Appendix A).

### 3.3. Risk-eSNPs were Enriched at Regulatory Regions

Given that risk-eSNPs influenced gene expression, we analyzed whether risk-eSNPs were enriched on regulatory or chromatin open regions. We overlapped risk-eSNPs with each histone or DNase mark of the craniofacial tissues and compared them with randomly generated SNPs that were matched by number, frequency, and TSS distance. We found that risk-eSNPs were statistically significantly enriched (~25–70 fold) in regions with active epigenetic modifications, including trimethylation of histone H3 at lysine 4 (H3K4me3) and trimethylation of histone H3 at lysine 36 (H3K36me3), but not in regions with the repressive modification trimethylation of histone H3 at lysine 9 (H3K9me3) (Figure 2B). These data indicated that risk-eSNPs were enriched in promoter or chromatin open regions of the craniofacial tissues.

Another feature of regulatory regions is that they are enriched in transcription factor binding sites. We next extended the analysis on binding transcription factors and found that risk-eSNPs were statistically significantly enriched in transcription factor binding sites, such as CHD1, GATA6, KDM4A, CTCF, TBP, E2F6, SP1, NANOG, YY1, and RAD21 (Figure 2C). Many of these transcription factors have been shown to be important in lip development or NSCL/P [22,23,24,25,26]. In addition, the eSNP-TF-eGene regulatory relationships for the top five TFs are listed in Appendix A. In summary, risk-eSNPs were enriched in regulatory regions, thus indicating the robustness of the approach.

### 3.4. Pathway Enrichment Analysis

GO and KEGG enrichment analyses were performed to explore the underlying risky biological processes and pathways. The GO enrichment analyses demonstrated that eGenes regulated by risk-eSNPs were mainly enriched in cell fate commitment, cellular response to retinoic acid, ERK1 and ERK2 cascade, neuron differentiation, and Wnt signaling pathways; they were also enriched in carcinoma, cancers, the pluripotency of stem cells, and mTOR signaling pathways in KEGG analyses (Figure 3).

### 3.5. Differential Expression Analysis

For the above 18 NSCL/P functional genes displaying eQTL signals, we analyzed mRNA expression levels in two gene expression microarray datasets (GSE42589 and GSE85748). In these two expression data sets, we found that 8 of the 18 effect genes were differentially expressed in at least one study (*p* < 0.05). For example, ZNF740 (*p* = 2.75 × 10^−2^) were upregulated in GSE85748. Meanwhile, UBL7 (*p* = 3.97 × 10^−3^), WNT9B (*p* = 3.66 × 10^−5^), and DLK1 (*p* = 7.14 × 10^−3^) were upregulated, while RAD54B (*p* = 1.69 × 10^−6^), CSK (*p* = 4.69 × 10^−2^), SPRY2 (*p* = 1.32 × 10^−4^), and TAF1B (*p* = 2.33 × 10^−2^) were downregulated in GSE42589 (Figure 4).

## 4. Discussion

In this study, we identified a number of SNP gene association pairs in lip tissues from patients with non-syndromic cleft lips with or without cleft palate (NSCL/P). By integrating nominal eSNPs with NSCL/P risk SNPs, we identified 243 risk-eSNPs and 18 susceptibility genes, which broadened our understanding of the genetic factors of NSCL/P.

Integrating functional genomics data to annotate specific sets of SNPs provided valuable information for understanding the underlying mechanisms of these SNPs [27]. In this study, we found that risk-eSNPs were enriched in open chromatin regions and transcription factors, which played critical roles in the development of NSCL/P. For example, abnormal neural crest migration was observed in Xenopus laevis embryos deficient in *Chd1* [22]. In addition, missense and copy number variants of *CHD1* contributed to craniofacial malformations, including cleft palate in human patients [23]. *GATA6*, which played an important role in the regulation of cellular differentiation and organogenesis during vertebrate development, was involved in the dexamethasone-inducing cleft palate [24]. Our study highlighted the importance of these regulatory elements and transcription factors in the development of NSCL/P.

A pathway enrichment analysis showed that the genes we identified were significantly enriched in classical developmental pathways, such as the Wnt signal pathway, which was a critical regulator of lip and palate formation [28,29]. Moreover, a KEGG analysis also found that many cancer-related pathways were enriched, such as breast cancer, gastric cancer, and basal cell cancer. Interestingly, several studies have reported the co-occurrence of NSCL/P and a variety of cancer types [30,31]. Our results further suggested that NSCL/P shared some genetic factors with cancers.

Among the eight differentially expressed genes, some are famous and known for their association with NSCL/P. Take *WNT9B* for instance, it belongs to the Wnt signaling pathway, which is a classic pathway in the development of the branchial arches, lip fusion, and normal morphogenesis of the face [32,33,34,35]. A Pbx-dependent Wnt-p63-Irf6 regulatory module in the midfacial ectoderm was established in a mouse line lacking Pbx genes in the cephalic ectoderm, and it was conserved within mammals. The abnormal expression of Pbx in the cephalic ectoderm led to a fully penetrant cleft lip with or without cleft palate (CL/P) and perturbed Wnt signaling, while ectopic Wnt ectodermal expression in Pbx mutants rescues the cleft. These findings indicated the significant role of adequate *WNT9B* expression in midface development and its association with NSCL/P [36,37].

Among some unfamiliar genes in NSCL/P, *TAF1B,* and *ZNF740* are transcription factors, which belong to the TATA-box binding protein-associated factors family and the zinc finger protein family, respectively. They involve in the universal transcription regulation process in cells [38,39,40] and play important roles in growth, aging, and responses to abiotic and biotic stresses. Despite their similar functions within their families, their independent roles and interactions with other family members or genes are still largely unveiled. In addition, their roles in NSCL/P still need to be validated in populations and verified by in vivo as well as in vitro experiments.

*DLK1*, delta-like non-canonical Notch ligand 1, is a member of the Notch signaling pathway. It has been found to decrease expression and increase differentiation as gestation proceeds, and its expression is restricted to a few tissues and progenitor cells in adults, but it would be re-expressed during disease and regeneration [41]. Here, we found that it was significantly upregulated in NSCL/P cases, which is in accordance with the previous findings and suggests its involvement in the occurrence of NSCL/P.

There were still some limitations in this study. Firstly, due to the limited sample size, some eGenes and eSNPs may be missed. Secondly, the technology principle of eQTL analysis, which mainly detected the genes with relatively high expression in the tissues, may lead to some undetected signals. Thirdly, the SNP array used for lip sample genotyping did not include all NSCL/P risk SNPs, which may also contribute to some variants missing in the integrative analysis. Finally, these lip tissues were sourced several weeks after the likely molecular events that led to the clefts happened. That means our findings cannot fully reflect the mechanism of the clefts. These problems will be overcome with the emergence of higher-quality data.

In conclusion, we generated a unique, lip-specific eQTL resource from 40 NSCL/P patients and identified multiple associations for NSCL/P risk loci. These findings provided new evidence for susceptibility genes of NSCL/P and new research directions for the future.

## Figures and Tables

**Figure 1 cells-11-03281-f001:**
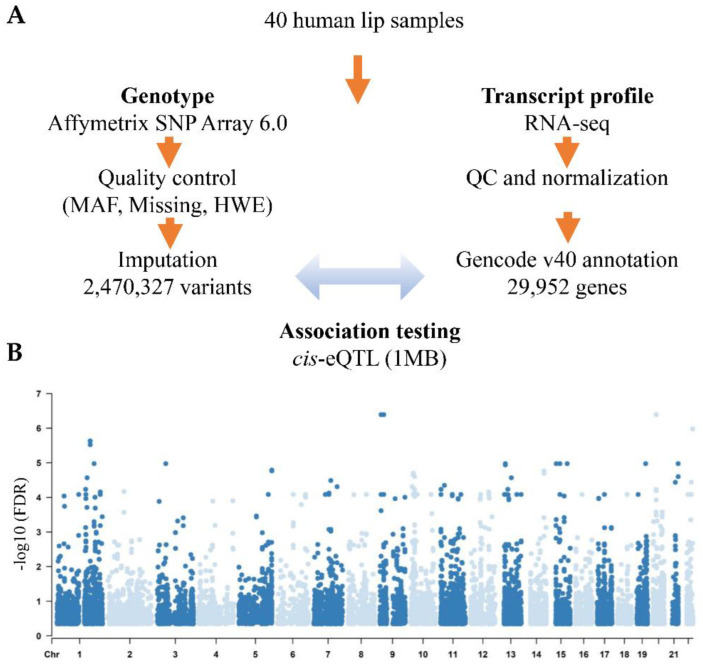
cis-eQTL analysis of lip tissues from NSCL/P patients. (**A**) Flowchart of 40 samples used for eQTL analysis, depicting the steps used to generate the genotype and gene expression data used in the eQTL analysis. (**B**) eQTL SNP-gene pairs across the genome. The x-axis shows the chromosome number, and the y-axis shows the -log_10_ (FDR) value.

**Figure 2 cells-11-03281-f002:**
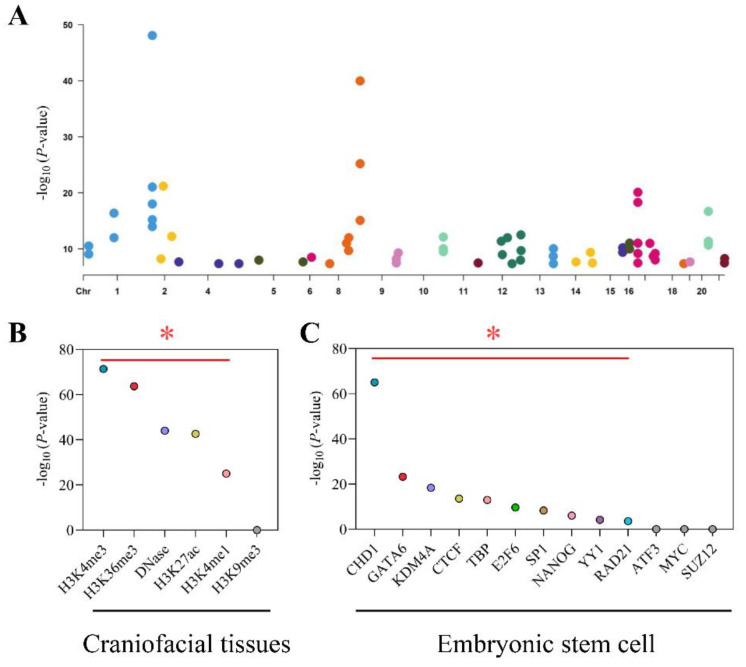
Identification and functional annotation of risk-eSNPs. (**A**) Significant NSCL/P risk SNPs across the genome. The x-axis shows the chromosome number, and the y-axis shows the -log_10_ (*p*) value. (**B**,**C**) The -log_10_ (*p*) value of eSNPs enrichment in histone marks and transcription factors, respectively. Marks with a significance level of *p*-value < 0.01 are highlighted above the red dotted line. * *p* < 0.01.

**Figure 3 cells-11-03281-f003:**
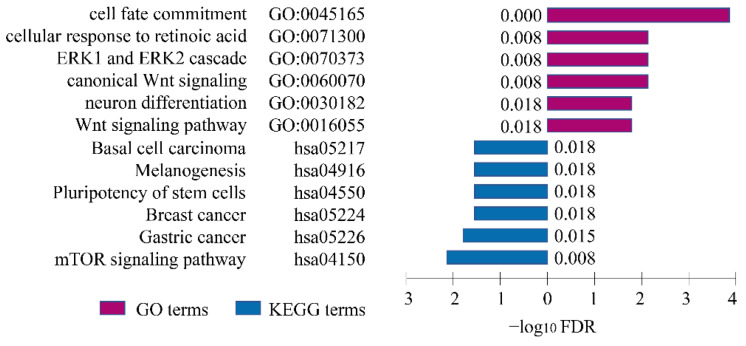
Pathway enrichment analyses. GO and KEGG pathway enrichment analyses for genes regulated by NSCL/P risk SNPs. Purple columns represent GO terms and blue columns represent KEGG terms.

**Figure 4 cells-11-03281-f004:**
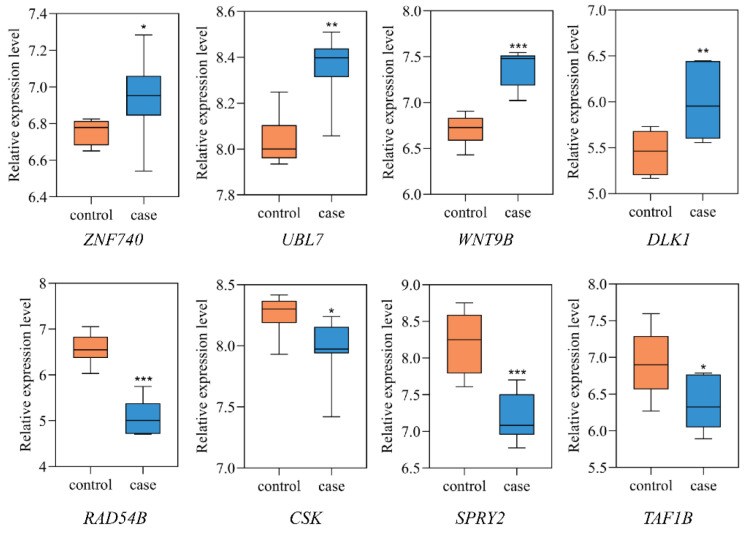
Eight differentially expressed genes between NSCL/P and healthy controls. The expression levels of differentially expressed genes in dental pulp stem cells (GSE42589) or orbicularis oris muscles (GSE85748) were compared between NSCL/P patients and healthy controls. * *p* < 0.05, ** *p* < 0.01, *** *p* < 0.001.

## Data Availability

The data presented in this study are available in insert article or Appendix A here.

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
