# Peer review of "Expression Quantitative Trait Locus Study of Non-Syndromic Cleft Lip with or without Cleft Palate GWAS Variants in Lip Tissues"

_cells, 2022, doi:10.3390/cells11203281_

Round 1

Reviewer 1 Report

Cleft lip with or without cleft palate is one of the most common craniofacial birth defects. In this manuscript, Li et al. generate a human lip-specific eQTL dataset from NSCL/P patients and integrate the eQTL SNPs with NSCL/P risk SNPs. These Risk-eSNPs and eGenes identified are valuable for future investigations. The experiments are technically sound and appropriately described despite some limitations as the authors discuss, e.g., the small sample size. Overall, the paper is well written. Below I have a few suggestions to improve the manuscript:

1.      For Transcription Factor Binding Site Enrichment Analysis (Section 2.5), the authors use ChIP-seq data in ESC (embryonic stem cell). However, transcription factor binding is dynamic and tissue specific. As the eQTL SNPs are identified from lip tissues, it is expected to use ChIP-seq data from lip tissues for this enrichment analysis. The authors should use more relevant ChIP-seq data or explain why the ESC data are used.

2.      In Discussion, Line 182, “In addition, missense and copy number variants of CHD1 contributed to craniofacial malformations including cleft palate in human patients.” References are missing. In Ref 22, Wyatt et al. reported a CLP case with a heterozygous deletion of CHD1 and RGMB, which could be supportive to this study. Are there other NSCL/P cases with variants of CHD1 reported?

3.      In this study, 243 risk-eSNPs are located within 250 kb of the 18 eGenes. The authors also identify that these risk-eSNPs are enriched in transcription factor binding sites. This may indicate that these eGenes are potential targets of the transcriptional factors such as Chd1 and GATA6. I am curious about which eGenes are potentially regulated by the specific transcriptional factors in the enrichment analysis. This would be interesting and could be discussed.

Reviewer 2 Report

This is well-presented work and the authors should address their main limitation, which is that lip tissues are sourced several weeks after the likely molecular events that led to clefts happened. That means that their findings may be completely unrelated to the causes of the clefts and just a reflection of the tissues functioning at that stage of life.
